# Development and Remodeling of Point-of-Care Ultrasound Education for Emergency Medicine Residents in Resource Limited Countries during the COVID-19 Pandemic

**Kamonwon Ienghong** [1] , **Lap Woon Cheung** [2,3], **Somsak Tiamkao** [4], **Vajarabhongsa Bhudhisawasdi** [1] **and Korakot Apiratwarakul** [1,*]

1   Department of Emergency Medicine, Faculty of Medicine, Khon Kaen University, Khon Kaen 40002, Thailand; kamonwan@kku.ac.th (K.I.); md221@kku.ac.th (V.B.)
2   Accident & Emergency Department, Princess Margaret Hospital, 2-10 Princess Margaret Hospital Road, Lai Chi Kok, Kowloon, Hong Kong; clw445@ha.org.ha
3   Emergency Medicine Unit, Li Ka Shing Faculty of Medicine, The University of Hong Kong, Pokfulam, Hong Kong
4   Department of Medicine, Faculty of Medicine, Khon Kaen University, Khon Kaen 40002, Thailand; somtia@kku.ac.th
*   Correspondence: korakot@kku.ac.th; Tel.: +66-0433-66869

**Abstract:** The administration of an accurate and effective POCUS course is a crucial tool in improving health education and thus the health care system in low- to middle-income countries. The development of the ultrasound curriculum in these countries during the pandemic era is a major challenge for medical educators. Therefore, this study aims to survey the learner experience after implementing the POCUS curriculum for first-year emergency medicine residents. All learners responded to the survey. Our results demonstrated that the ultrasound rotation and our ultra-sound learning materials were useful tools which showed a positive impact on POCUS knowledge for our learners. However, some obstacles of POCUS learning were identified to assist in closing faculty development gaps, including the availability of handheld devices, as well as the re-modeling of the ultrasound rotation course, which should be managed according to the feedback we received. This study demonstrated a clear need for constant updates in higher education, medical program development, accuracy of local learning materials, and the explosion of virtual and online learning platforms during this decade.

**Keywords:** medical education; teaching; ultrasound; emergency medicine; COVID-19; sustainability; program development

## 1. Introduction

A sustainable healthcare system is defined by the World Health Organization (WHO) as a system that improves, maintains or restores health, while minimizing negative impacts on the environment and leveraging opportunities to restore and improve it, to the benefit of the health and well-being of current and future generations. The development of novel knowledge and the technology referred to as the point-of-care ultrasound (POCUS) has shown great potential in providing sustainable health care.

For the past two decades, the primary tools for bedside diagnosis are the integration of a patient's history and a physical examination. This is especially true in resource-limited countries. After having been introduced as a bedside tool in the practice of emergency medicine, the POCUS is proving to play an increasingly crucial role in diagnosis and treatment in emergency patients globally [1–3]. POCUS offers a relatively low-cost tool to identify pathology, provide a rapid diagnosis, allow for prompt treatment, and to assist in performing bedside procedures for life-threatening issues [4–7] in resource-limited settings where other imaging modalities are cost-prohibitive. Thus, the utilization of POCUS is widely recognized to be of great benefit.

However, there are some common POCUS errors or pitfalls for diagnosis in novice POCUS usage. Thus, to be professional in performing POCUS, this novel knowledge needs to be delivered, in addition to training, in a formal way, through a trainer or an established formal curriculum. Recent innovations in POCUS training and education have surged and should now be a key element in medical training.

In 1994, the POCUS curriculum for training emergency physicians was developed and implemented in the United States and Canada [8]. In 2001, the American College of Emergency Physicians (ACEP) also announced the first set of emergency ultrasound guidelines [9]. There are established POCUS curricula available in many developed nations, e.g., North America, Europe, Australasia, and South Africa [10–12]. However, in many resource–limited countries, especially in Southeast Asia, there is no formal curriculum training in emergency medicine resident programs due to the small number of POCUS specialists. In the context of POCUS resources being limited in Thailand, (1) there is no official POCUS learning in medical curriculum, (2) in many hospitals, there are no ultrasound machines in the emergency departments, and (3) there are less than 10 certified POCUS specialists in Thailand.

About 10 years ago, POCUS was introduced to the Emergency Department in Thailand by The World Interactive Network Focused on Critical Ultrasound (WINFOCUS). This organization collaborated with the King Chulalongkorn Memorial hospital to establish the WINFOCUS Thailand. After that, there was a certified POCUS training workshop for Thai emergency physicians. In 2018, the POCUS education for emergency medicine residents was established by the Thai College of Emergency Physicians (TCEP). Initially, the TCEP held an annual formal procedure workshop called "TCEP Resuscitative Procedure Courses", which had multiple sessions of key procedures for resuscitation when using POCUS. The ability to perform emergency POCUS was included in the Thailand EM (Emergency Medicine) milestones in 2018 by the TCEP EM training board. EM residents are mandated to perform POCUS examinations and undergo an evaluation of POCUS skills according to entrustable performance activities (EPA) using the direct observation of emergency procedural skill assessment (DOPs) before taking the board examination.

Our institution (the Department of Emergency Medicine, Faculty of Medicine, Khon Kaen University) is the medical academic center in the Northeast region of Thailand. We have been active in training emergency physicians across Thailand since 2007. Training in POCUS is a key component of EM. Thus, the ultrasound curriculum was first implemented in July 2019. Our faculty arranged POCUS training for first year emergency medicine residents in the 2019–2020 academic year and the 2020–2021 academic year. However, in the 2020–2021 academic year, the POCUS education program was remodeled into an online platform due to the COVID-19 pandemic. Alongside the development of POCUS education in our institution, the purpose of this study was to define the quality of the proprietary training in the field of POCUS by using the survey affiliated with our institution's postgraduate EM training programs to assess the opinions of learners about our ultrasound curriculum and the learning materials used during this challenging time.

## 2. Materials and Methods

### 2.1. Study Design and Setting

A cross-sectional survey study was conducted at Srinagarind Hospital, Department of Emergency Medicine, Thailand from July 2019 to July 2021. This hospital has an average of 70,000 emergency patients visiting per year.

### 2.2. Study Participants

Participants were all first-year emergency medicine residents who attended the ultrasound rotation during the 2019–2021 academic year. Participants who did not attend this rotation were excluded from this study.

### 2.3. Data Collection

A 10–question survey was conducted (Appendix A). Respondents were asked to determine the experience of POCUS learning by using the 5-Likert scale after finishing the ultrasound rotation in 1 week. A validated questionnaire was used and emailed to all participants. An e-mail containing a link to the survey was sent to each participant, which was personalized for each user, to track responses and e-mail non-responders. However, data collected were anonymous. Non-responders were contacted two times over 2 weeks.

### 2.4. Study Size

The residents, who were in the ultrasound rotation at the Department of Emergency Medicine at Khon Kaen University's Srinagarind Hospital from July 2019 to July 2021, were enrolled in this study. Hence, based on a questionnaire study, we estimate that there were around eighteen residents who would meet our inclusion criteria. Assuming a participation rate of 60% [13], we expected to include at least eleven residents in this study. Therefore, the study group was comprised of first-year emergency medicine residents who attended the first POCUS curriculum developed by our faculty during 2019–2021 academic year and who had responded to our survey questionnaire.

### 2.5. Development and Implementation of POCUS Education and POCUS Learning Tools

The ultrasound curriculum (Table 1) was a two-week rotation which consisted of the journal club (reviewing of the latest ultrasound journal), the process of reviewing the ultrasound images collected by the learner, and the didactic lectures, as well as bedside ultrasound learning with real patients, which was conducted by a supervisor, who was a POCUS specialist.

**Table 1.** The ultrasound curriculum for first year emergency medicine resident before and after pandemic year.

| Core Emergency Ultrasound Application | Duration | | Description |
|---|---|---|---|
| **Before Pandemic (2019–2020 Academic Year)** | | | |
| 1. The journal club | 1 session, 3 h | - | 1 ultrasound paper per section |
| 2. The review of ultrasound image | 1 session, 3 h | - | Review of all ultrasound images collected from the students for 2 weeks |
| 3. Didactic lecture | 3 h per week | - | Didactic lecture included all 10 contents; cardiac, lung, abdomen, aorta, deep venous thrombosis, soft tissue and musculoskeletal, ocular, kidney and urinary system, obstetric and gynecologic system, procedural guidance, ultrasound protocols—FAST and EFAST, RUSH, BLUE, CASA protocol) |
| 4. Bedside ultrasound | 9 h per week | -<br>-<br>- | Bedside ultrasound teaching with real patients<br>Demonstrated POCUS performing by POCUS specialist<br>Student performed ultrasound under POCUS specialist supervision |
| **After Pandemic (2020–2021 Academic Year)** | | | |
| 1. The journal club | 1 session, 3 h | - | 1 ultrasound paper per section by using Video conference via Zoom |
| 2. The review of ultrasound image | 1 session, 3 h | -<br><br>-<br><br><br>- | Review of all ultrasound images collected from the students during 2 weeks by using Video conference via Zoom<br>Review of online image banks (e.g., The POCUS Atlas, www.ultrasoundcases.info (accessed on 14 September 2021), 123sonography.com (accessed on 14 September 2021))<br>Teaching images virtual library sharing via Google Drive |
| 3. Didactic lecture | 3 h per week | -<br>- | Video conference via Zoom<br>Self-directed learning with online modules (10 contents) |
| 4. Bedside ultrasound | 9 h per week | -<br>- | Learners used each other as mannequins or used mock patients<br>Tele-ultrasound modalities (FaceTime, video call) |

*2.6. Remodeling of POCUS Education*

In the 2020–2021 academic year, we were faced with the relentless threat of a resurgent Coronavirus pandemic; thus, we had changed all formerly in person POCUS training courses to online learning via various platforms, including the Zoom and LINE call applications. Moreover, bedside ultrasound teaching with a real patient was not provided that year (Table 1).

Thus, the difference of our POCUS curriculum before and after pandemic era was (1) the platform of learning was onsite before the pandemic and 100% online after the pandemic, (2) the number of learning hours was the same, (3) in terms of bedside learning, we used mock patients instead of real patients. The impact of the remodeling course was (1) students can learn from our online platform any time and repeatedly, (2) Thai students usually fear to ask questions if the learning platform is onsite; therefore, when students learn by online platform, they can ask more questions via the chat box.

Objectives of the curriculum: by the end of this rotation, learners will be able to demonstrate:

1. Competency in the following POCUS applications: identifying the presence of intraperitoneal free fluid, presence of pericardial fluid, measurement of IVC and aorta diameter, presence of cardiac motion, assessment of fluid status by using cardiac, lung and IVC ultrasound, confirmation of intrauterine gestation, hydronephrosis, pneumothorax and gall stone), and facilitation of vascular access.

2. Competency in basic ultrasound protocols (FAST, EFAST, RUSH, BLUE and CASA protocols).

3. The ability to provide good image acquisition and good image interpretation in core POCUS organs (cardiac, lung, IVC and aorta).

4. Competency in clinical integration of ultrasound findings.

In terms of the materials for learning POCUS, there is no standard textbook written by the POCUS specialists in Thailand. Our faculty developed "The Practical Ultrasound Flashcards with Augmented Reality" (Figure 1). The set was designed in the form of 125 pages of cards that contain twenty ultrasound clips and eleven fundamental aspects of POCUS knowledge, which are included in all of the didactic lectures. The learner uses these tools in the ultrasound rotation.

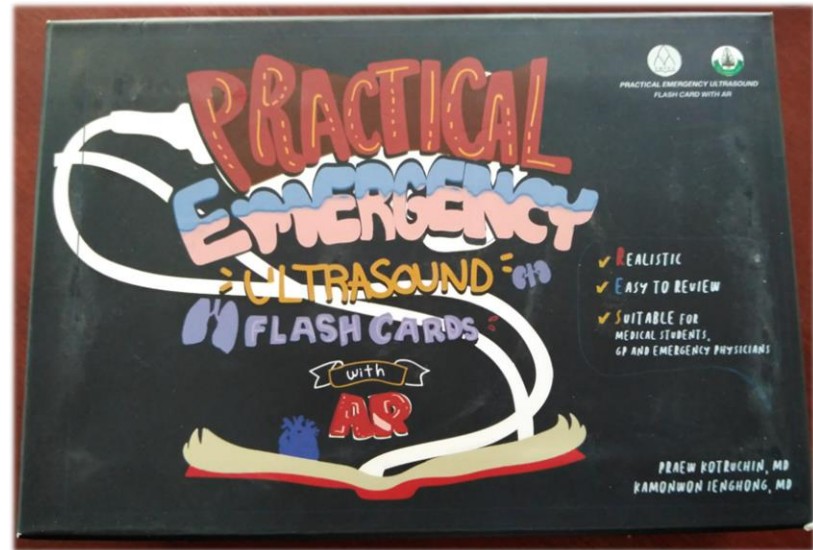

**Figure 1.** POCUS learning tool: "The Practical Ultrasound Flashcards with Augmented Reality".

In terms of the ultrasound phantom, our faculty developed the ultrasound phantom for practicing ultrasound guided peripheral intravenous assessments (UGPIV), due to the high cost of the commercial model for practicing this skill. The model was created based on a study by Chao [14] and the phantom was created using gelatin powder, fiber powder,

and a latex tube (Figure 2). We taught our residents to create this model and then they used it to practice their UGPIV skills.

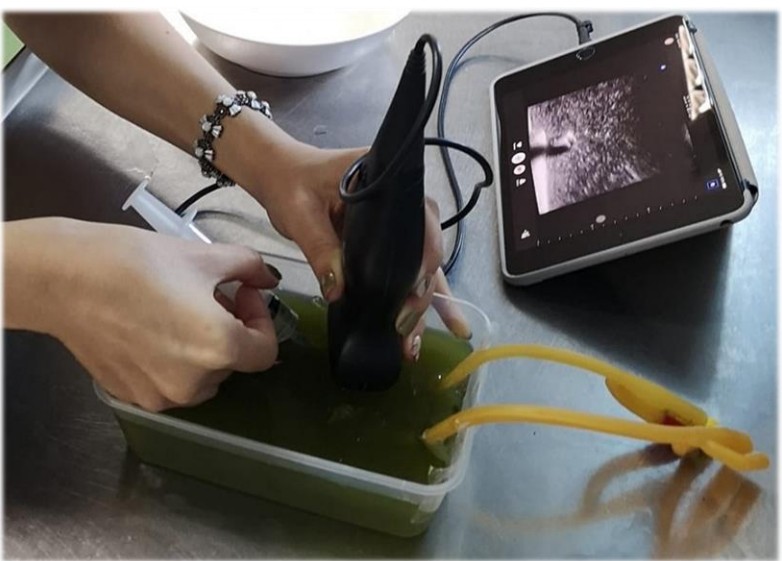

**Figure 2.** The ultrasound simulation model for practicing UGPIV.

In terms of the ultrasound machine, we provided 2 ultrasound machines in the emergency department for performing clinical practice with the patient and also for POCUS learning. The ultrasound machine was an M9 (Mindray, Shenzhen, China) and Sonosite M turbo (Fujifilm, Washington DC, USA) equipped with a curved array probe, phased array probe, and linear probe. Moreover, we provided 1 handheld ultrasound named Butterfly IQ.

*2.7. Ethical Considerations*

Ethical approval was provided by the Khon Kaen University Ethics Committee for Human Research (HE641406). The written informed consent was obtained from each participant before the study.

**3. Results**

A total of eighteen EM residents attended the entire ultrasound rotation between July 2019 and July 2021. All of those responded to the survey (100%). All of the respondents (100%) had POCUS experience before attending this rotation. Most respondents reported the number of POCUS examinations in this rotation to be about 31–40 times and considered this to be useful in improving POCUS knowledge. The most useful learning activities were the bedside ultrasound teaching and the review of the ultrasound image, respectively. All respondents rated "Very Useful" to the ultrasound flashcard as the most useful of the ultrasound learning tool (Figure 3). Regarding the suitable time management of each activity, 94% of the respondents reported "Very suitable" for the bedside ultrasound section. Regarding the ultrasound devices, 83% of the respondents identified the Mindray M9 as easy to use in the program.

The POCUS examination type that respondents practiced in this class was displayed in Figure 4. Most of students performed cardiac and lung ultrasounds most of the time in this rotation. The perceived barrier to learning POCUS was reported in Figure 5. Most of the students rated ER overcrowding as a "Very large barrier" to learning POCUS. A total of 98% of respondents reported that it is "important" or "very important" to have a POCUS curriculum in an emergency medicine residents' training program.

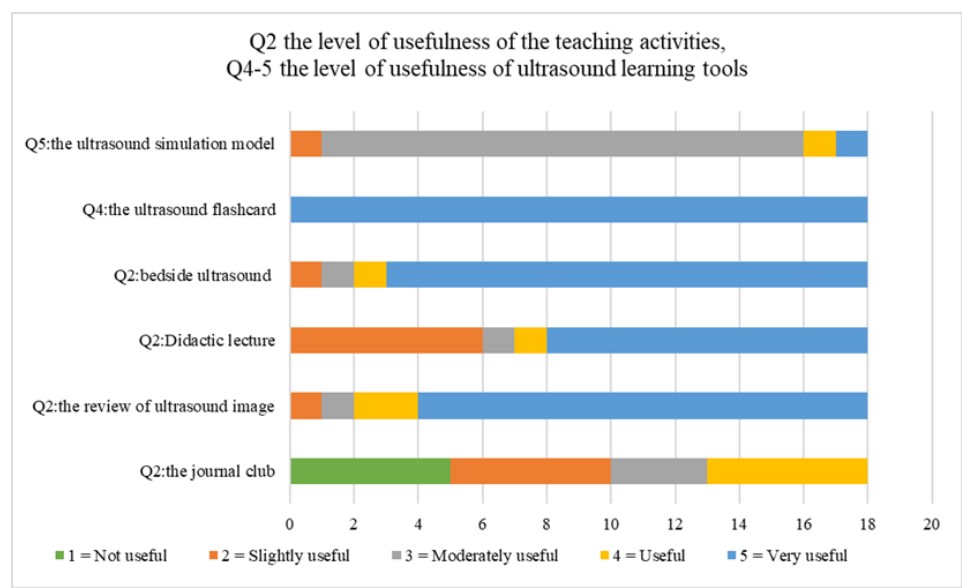

**Figure 3.** Comparison of the usefulness level in each of the teaching activities and ultrasound learning tools.

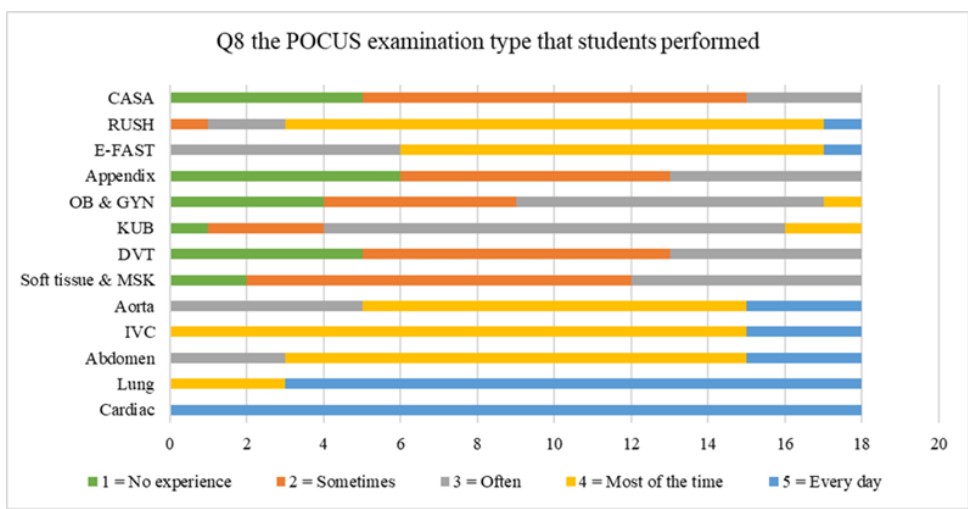

**Figure 4.** Comparison of the frequency of the POCUS examination types that students performed.

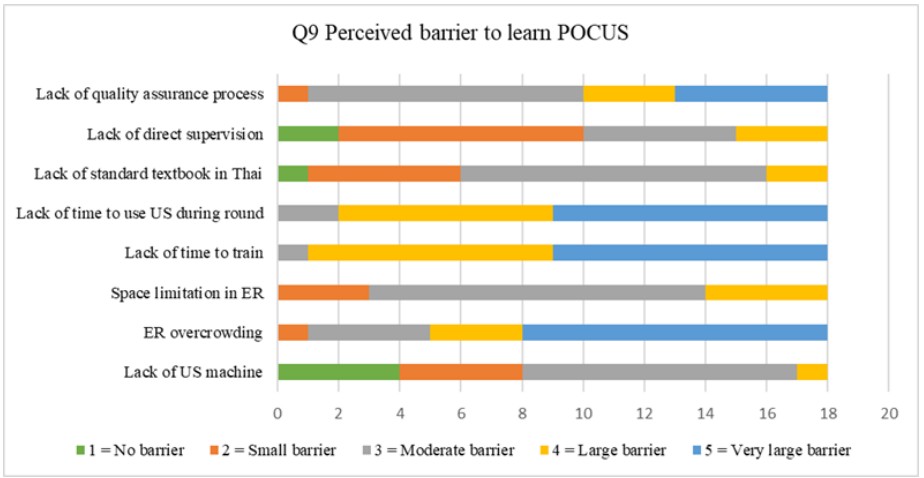

**Figure 5.** Rating scale of the perceived barrier to learning POCUS.

## 4. Discussion

The aim of this study was to explore the learners' opinions regarding the ultrasound rotation, which included teaching activities, timing and length of training, ultrasound learning tools, the barrier to learning POCUS and the learners' points of view to POCUS education.

### 4.1. Training Program

In terms of education methods, there are a lot of teaching methods utilized in the instruction of POCUS; for example, didactic lectures, formal quizzes or quizzes in the form of games, live model demonstration, simulation training, and direct patient scanning. The utility of each method was different [15]. However, the combination of varieties of teaching methods can be the most effective way to achieve the objective of the curriculum. In this study, we provided four types of teaching methods for learners, including didactic lecture, bedside ultrasound teaching, image review and the journal club.

Our results indicated that most learners identified the teaching activities that can best improve POCUS knowledge were the bedside ultrasound teaching and the process of reviewing ultrasound images with a supervisor in both academic years, which is consistent with many studies [16–19]. These sections allowed learners to improve the competency of image acquisition and image interpretation with real time feedback from the supervisor. Moreover, the learners can cooperate to synthesize the ultrasound findings along with the patient's chief complaint and symptoms into clinical decision making.

Regarding the hands-on section, in the first academic year of the ultrasound rotation, we provided bedside ultrasound teaching by performing POCUS examinations with real patients. However, in the second academic year, the global community faced the COVID-19 pandemic, and we did not dare to arrange this section due to safety concerns. The EM residents were mandated to perform POCUS examinations with the mock patients or used each other as mannequins. Because of this, the supervisor was not able to demonstrate the real pathologic sonographic features. This was the issue that we were most concerned about, so we suggested some online resources, such as www.ultrasoundcases.info (accessed on 14 September 2021) and https://123sonography.com/point-of-care-ultrasound (accessed on 14 September 2021) for additional supplemental learning materials.

The next activity that learners preferred was the didactic lecture. This study designed the didactic lecture adapted from the American College of Emergency Physicians (ACEP) which consisted of 12 core ultrasound applications [12]. The content of the lecture included all core knowledge. However, the delivery vehicle of the lectures was different in the second academic year, when we were forced to switch to the online platform. In a study comparing the traditional lecture with e-learning, the authors demonstrated there was no evidence of a difference between the two groups in terms of learning capacity or satisfaction outcomes [20]. Moreover, utilization of a video-based conferencing platform, such as Zoom, offers the ability to transfer knowledge in real time as well as record didactics for future viewing [21,22]. Thus, learners can continue to learn in the traditional scheduled onsite methods while also being able to learn on their own time.

The teaching activity that the learners rated of the least value according to scores was the journal club. In many medical curricula, there was no section or only a little section of the journal club in the POCUS education for medical students and residency programs, except for in the fellowship program. Most academic settings provided the journal club to encourage learners to develop critical appraisal skills and utilize current best practice [23]. However, for first year emergency medicine residents, learners aimed to gain more experience in POCUS practice skills and basic POCUS knowledge.

In terms of POCUS examination types that learners had practiced in this rotation, most learners performed cardiac and lung ultrasounds every day. The next most frequent was the IVC examination, which most learners performed. These are comprehensive ultrasound examinations for the assessment of patients with cardiopulmonary complaints and undifferentiated shock, which is vital for novice users to recognize, gain practice

in, and gain as much experience as possible to rapidly improve their skill. Interestingly, most learners reported they had no experience of OB-GYN ultrasounds nor those for the Appendix. This may be due to (1) a negligible number of OB-GYN patients in the ER, (2) most obstetricians prefer transvaginal ultrasounds that are not provided in the ER, or (3) in cases of appendicitis, the surgeon prefers formal ultrasounds from a radiologist.

As for the number of scans conducted by students, our learners reported their number of scans in this rotation were 31–40 times in 2 weeks. Current ACEP guidelines suggest a minimum of 25 ultrasound scans for each type of scan performed in the practice of ultrasound scanning skill training [12]. It should be noted that during our research we did not know the exact number of scans per examination due to the fact that we were not recording. Thus, we cannot claim that a specific number can improve the learners' POCUS skills.

### 4.2. Timing and Length of Training

The design of a POCUS training program was a challenging task for our faculty due to many limitations, including (1) there was only one supervisor who is a POCUS specialist, (2) the already dense nature of the emergency medicine curriculum, (3) the limitation of the time in which learners have to participate in this rotation, and (4) a limitation of educational resources for learning POCUS in Thailand. Therefore, we arranged the ultrasound rotation for only 2 weeks for first year residents. This may affect the learners' practical skill due to the insufficiency of practice time and retention of the knowledge. In a study about the retention of knowledge, it was found that the addition of a longitudinal component to ultrasound education may result in improved knowledge retention [24]

Our results clearly showed the length of training in bedside ultrasound teaching was very suitable for residents. In our experience, there is something that is more important to consider than the length of learning time and that is the limitation of the number of learners per supervisor per ultrasound machine and the number of POCUS examinations performed by learners in each class; this is consistent with the findings of other studies [25–29].

In terms of the optimal number of total hours for the ultrasound rotation, there is quite a range from as little as 4 h to as long as one month [30,31]. However, in a study assessing the residents' competency, the study demonstrated that a 2-week rotation immediately improved short term ultrasound knowledge among the EM residents [32].

### 4.3. Ultrasound Learning Tools

To achieve the 2030 Sustainable Development Goals (SDGs) for health—specifically, goal 3 (Good Health and Well-Being) and goal 4 (Quality Education)—effective medical education development is a crucial consideration. However, in low- and middle-income countries, the greatest limitation of health care and education is the budget. There are many varieties of POCUS teaching tools that are currently used commercially, such as Sonosim, which is an ultrasound training solution that provide hands-on online simulators, Insimo, which is another online platform ultrasound simulator, and iTeachU, which is a POCUS eLearning package, etc. The problem is that they are not affordable in lower income countries; for example, the popular commercial phantom, the Blue Phantom's Branched 2 Vessel Ultrasound Training Block Model, has a price of around USD 500 per model. Thus, the development of their own tools with the incorporation of local ingredients may provide great help to the learners to access this knowledge and skill set. The first subject is the Ultrasound Flashcard using Augmented reality. This was the first POCUS education tool to be developed in our faculty, as well as in all of Thailand. In the pilot phase, our faculty implemented this tool to assist in the training of medical students who attended the emergency medicine rotation. There was clear evidence that medical students who used this tool to learn POCUS left with better knowledge scores than the others who attended the standard ultrasound training course only [33]. This is consistent with our findings, which showed that most learners rated this tool as "very useful". The second subject is the ultrasound phantom, which our faculty created for the purpose of the simulated

practice of UGPIV with the supplement of some local ingredients. The advantages are low cost, easy to make, and reproducible, which is consistent with other studies [34–36]. However, our learners reported it as only "moderately useful". Most learners prefer the commercial phantom due to the firm, more life-like consistency, and the echogenicity of the commercial phantom.

*4.4. Perceived Barrier to Learning POCUS*

The main barriers in this rotation were the lack of time and a sizable number of patients in the ER, which is consistent with other studies [28,37]. Our learners reported they had little time to use the ultrasound machine when they wanted to practice scanning for educational purposes. In addition, there were too few ultrasound machines in the ER, which were used both for clinical practice and for education and the ER treatments. Again, overcrowding is the main problem in low- to middle-income countries. As the pressures continue to mount, the solution may simply be the availability of handheld ultrasound devices, which are small, affordable, feasible, and easy to access, which is consistent with other studies [38–43]. Finally, the majority of learners reported that they felt the time to study in this rotation was too short.

The limitations of this study were (1) the small number of the participants and the single intuitional study that cannot represent all the POCUS learners experience, (2) this study used self-reported data; therefore, there were some biases, including selective memory and (3) this study was not designed to demonstrate the knowledge improvement of the learners and the retention of knowledge of the learners. This study focused on POCUS curriculum in first year emergency medicine residents, which may not be generalizable to other specialties. All outcomes reflect the EM residents' perceptions of the learning activities and POCUS learning materials in our programs, which are potential sources of error. The current study only surveyed program residents; thus, no comparative of the statistic was provided. However, the strength of this study included that the study was the first survey about the POCUS learning in Thailand, which was high in response rate.

## 5. Future Work

POCUS education in low- to middle-income countries is still shrouded in mystery. We know that due to many limitations, namely budgetary ones, coupled with the need for expensive equipment and courses, that many of these programs are inadequate. One of the potential avenues to developing quality POCUS education programs is by creating their own POCUS learning materials. The limitations put upon academic and medical institutions by the pandemic must be dealt with creatively; materials should be developed in accessible formats, which include eLearning or the availability of other online materials. The main benefit of eLearning is providing the POCUS knowledge to the many learners who currently struggle with a limited number of POCUS specialists. Moreover, to provide a more substantive curriculum evaluation based on learning outcomes, our future work will focus on the more direct and long-term measures of the student performance, which are necessary in a competent EM practitioner.

## 6. Conclusions

Further development of POCUS education through the implementation of the ultrasound rotation in emergency medicine residents' curricula is the first step to enhance proper POCUS usage in Thailand. In our study, we developed an ultrasound curriculum and ultrasound learning materials, which included the ultrasound flashcards and the ultrasound phantom. With the arrival of the COVID-19 pandemic, the global POCUS education was disrupted in a major way. Remodeling of this course was necessarily developed. The virtual or online platform plays a crucial role for conducting POCUS training and making it accessible to learners. Our results demonstrated that all academic activities employed and the ultrasound learning materials were useful in improving POCUS knowledge. However, we found some obstacles in POCUS learning that needed to be improved, mainly

the length of time of training. We believe that adopting this POCUS education and training method can lead directly to more effective treatment and be of great benefit to both future physicians and their patients.

**Author Contributions:** Conceptualization, K.A. and K.I.; methodology, K.A.; validation, K.A. and L.W.C.; formal analysis, K.A. and K.I.; investigation, K.A. and K.I.; resources, S.T. and V.B.; data curation, K.A. and K.I.; writing—original draft preparation, K.A. and K.I.; writing—review and editing, K.A., K.I. and L.W.C.; visualization, K.A.; supervision, S.T. and V.B.; project administration, K.A. and K.I. All authors have read and agreed to the published version of the manuscript.

**Funding:** This research was funded by the Research and Graduate Studies, Khon Kaen University, Thailand. (RP64-7/002).

**Institutional Review Board Statement:** The study was conducted according to the guidelines of the Declaration of Helsinki and approved by Ethics Committee of Khon Kaen University (HE641406).

**Informed Consent Statement:** Informed consent was obtained from all subjects involved in the study.

**Data Availability Statement:** Not applicable.

**Acknowledgments:** The authors would like to express our sincere gratitude to Josh Macknick for acting as an English consultant.

**Conflicts of Interest:** The authors declare no conflict of interest.

## Appendix A

**Table A1.** Validated questionnaire of the learning experience of POCUS in the ultrasound rotation.

| | | 1<br>Not Useful | 2<br>Slightly<br>Useful | 3<br>Moderately<br>Useful | 4<br>Useful | 5<br>Very<br>Useful | N/A |
|---|---|---|---|---|---|---|---|
| Q1 | 1. Gender | | | | | | |
| | 2. Age | | | | | | |
| | 3. POCUS Experience | | Yes | | | No | |
| Q2 | How useful do you think the following teaching activities are to help to improve POCUS knowledge? | 1<br>Not Useful | 2<br>Slightly<br>Useful | 3<br>Moderately<br>Useful | 4<br>Useful | 5<br>Very<br>Useful | N/A |
| | 1.The journal club | | | | | | |
| | 2. The review of ultrasound images | | | | | | |
| | 3. Didactic lectures | | | | | | |
| | 4. Bedside ultrasounds | | | | | | |
| Q3 | How do you think the duration of time used in the following activities is for improving your POCUS knowledge? | 1<br>Not Suitable | 2<br>Slightly<br>Suitable | 3<br>Moderately<br>Suitable | 4<br>Suitable | 5<br>Very<br>Suitable | N/A |
| | 1. The journal club<br>(1 session, 3 h) | | | | | | |
| | 2. The review of ultrasound images<br>(1 session, 3 h) | | | | | | |
| | 3. Didactic lectures<br>(3 h per week) | | | | | | |
| | 4. Bedside ultrasounds<br>(9 h per week) | | | | | | |
| Q4 | How useful do you think the ultrasound flashcards are in helping to improve POCUS knowledge? | 1<br>Not Useful | 2<br>Slightly<br>Useful | 3<br>Moderately<br>Useful | 4<br>Useful | 5<br>Very<br>Useful | N/A |
| Q5 | How useful do you think the ultrasound simulation model is to help to improve POCUS skills? | 1<br>Not Useful | 2<br>Slightly<br>Useful | 3<br>Moderately<br>Useful | 4<br>Useful | 5<br>Very<br>Useful | N/A |

**Table A1.** *Cont.*

| | | 1 Very Unsatisfied | 2 Unsatisfied | 3 Neutral | 4 Satisfied | 5 Very Satisfied | N/A |
|---|---|---|---|---|---|---|---|
| Q6 | Ease of use of the ultrasound devices used in this education program? | | | | | | |
| | 1. Mindray M9 | | | | | | |
| | 2. Sonosite M turbo | | | | | | |
| | 3. Butterfly iQ | | | | | | |
| Q7 | Number of independent POCUS examinations performed in this rotation | 0–10 | 11–20 | 21–30 | 31–40 | >41 | N/A |
| | Type of POCUS examination that the student practiced in this class | 1 No Experience (0%) | 2 Sometimes (20%) | 3 Often (50%) | 4 Most of the Time (80%) | 5 Every Day (100%) | N/A |
| | 1. Cardiac | | | | | | |
| | 2. Lung | | | | | | |
| | 3. Abdomen (liver, gallbladder, ascites) | | | | | | |
| | 4. Inferior vena cava (IVC) | | | | | | |
| Q8 | 5. Aorta | | | | | | |
| | 6. Soft tissue and musculoskeletal (MSK) | | | | | | |
| | 7. Deep Venous thrombosis (DVT) | | | | | | |
| | 8.Kidney and Urinary bladder (KUB) | | | | | | |
| | 9. Obstetrics and Gynecology | | | | | | |
| | 10. Appendix | | | | | | |
| | 11. E-FAST | | | | | | |
| | 12. RUSH protocol | | | | | | |
| | 13. CASA protocol | | | | | | |
| Q9 | Perceived barrier to learn POCUS | 1 No Barrier | 2 Small Barrier | 3 Moderate Barrier | 4 Large Barrier | 5 Very Large Barrier | N/A |
| | 1. Lack of ultrasound machine for learning | | | | | | |
| | 2. ER overcrowding | | | | | | |
| | 3. Space limitations in ER | | | | | | |
| | 4. Lack of time to use the ultrasound during round | | | | | | |
| | 5. Lack of time to train | | | | | | |
| | 6. Lack of a standard POCUS textbook in the Thai language | | | | | | |
| | 7. Lack of direct supervision | | | | | | |
| | 8. Lack of quality assurance process | | | | | | |
| Q10 | How important is it to implement a POCUS education section in the Emergency Medicine residency training program? | 1 Not at All Important | 2 Slightly Important | 3 Important | 4 Very Important | 5 Extremely Important | N/A |

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
