# Peer review of "Development and Remodeling of Point-of-Care Ultrasound Education for Emergency Medicine Residents in Resource Limited Countries during the COVID-19 Pandemic"

_tomography, doi:10.3390/tomography7040060_

Round 1
Reviewer 1 Report
The authors showed us a creative POCUS teaching tool: flashcard and its positive impacts on education. They also provided us better educational strategies for beginners.
The authors need to define more about "resource-limited" in this study
The remodeling was an adaption in the pandemic era. The information about the participants in different periods was insufficient. Meanwhile, the authors need to describe more about the difference and impact on the development and remodeling period.
Contents in table 1 & 2 should be integrated into one table.
Figures 3,4 and 5 should be re-constructed for better understanding.
In a rotation for only 2 weeks, it is difficult cover up to 13 POCUS contents and protocols in a total 6 hours didactic lectures. Not to mention the insufficiency of practice time.
The machines provided in this study were M9 & M turbo. Q6 (3) is Butterfly iQ ?
Author Response
Thanks for your comments on this manuscript.
I have already done to correct this manuscript as you suggest. Please see the attachment below

Reviewer 2 Report
Thank you for the opportunity to review the manuscript. The work concerns the important issue of education in the field of POCUS.
The authors' intentions and commitment to the project are commendable. However, the presented work has many fatal errors and has low scientific value.
1) In the introduction, the authors wrote that "the aim of the study was the questionnaire (...)". The purpose of the research study MUST NOT BE A SURVEY. The aim of the study may be, for example, to "define the quality of the proprietary training in the field of POCUS".
2) My basic concern is the methodology which only includes an anonymous questionnaire with ten question categories. It does not contain any sociodemographic data of participants, as well as scores on the effectiveness of the course (e.g. credits, exam, achieved learning outcomes). In my opinion, a purely subjective survey has very little scientific value. There is no evaluation of the effectiveness of the training provided here.
3) No statistics, no comparisons whatsoever.
4) A small group of participants. Describe if it was representative.
5) Lack of opinions from residents regarding the training program (whether the time and topics were sufficient, proposed changes, etc.)
6) The authors assumed that changing the mode to remote in learning POCUS makes no difference. Can authors provide resident grades for each course module in the results? The study should definitely show that changing the course to the online form can be comparable to the stationary course in the case of POCUS training.
7) The discussion describes "Perceived barrier to learn POCUS", which includes the opinions of the course participants, and not the limitations of the research (in the opinion of the authors). This should be corrected. Participants' opinions should be posted in the RESULTS section.
8) 41 references were presented, of which only half (22) can be considered new sources (not more than 3 years). In the field of POCUS, it is worth reaching for the latest sources, as it is a dynamically developing issue.
I recommend adding example items:
a) Wong, C. K .; Hai, J .; Chan, K. Y. E .; Un, K. C .; Zhou, M .; Huang, D .; et al. W. Point-of-care ultrasound augments physical examination learning by undergraduate medical students. Postgraduate Medical Journal, 2021,97 (1143), 10-15.
[POCUS learning efficiency]
b) Wejnarski, A .; Gajek Villebæk, P.A .; LeszczyĹ„ski, P.K. Prospective evaluation of interactive project of Emergency Medicine Exam with the use of multimedia computer devices. Crit Care Innov, 2018, 1 (1), 1-15.
[effectiveness of computer forms of teaching and examination]
Author Response
Thanks for your comment on our manuscript.
I have already corrected the manuscript as your recommended.
Please see the attached file below

Round 2
Reviewer 1 Report
The authors responded well to all the advice.
They also provided well prepared tables and figures for readers.
Author Response
Dear editor and reviewer,
Thanks for your valuable time in response to our manuscript.
Sincerely,
Kamonwon Ienghong
Reviewer 2 Report
The authors took into account only some of the reviewer's comments.
I still believe that the lack of an appropriate research methodology (description of the studied groups, statistics, substantive evaluation) significantly reduces the scientific value of the work.
Author Response
Dear editor and reviewer,
We appreciate your precious time in reviewing our paper. We tried our best to address every comment. We hope our manuscript after careful revision meets your standard. Below we provided point-by-point responses. All modifications have been marked up using the "track changes" function.
Please see the attached file below.
Best regards,
Kamonwon Ienghong
